# Morbidity of Returning Travelers Seen in Community Urgent Care Centers throughout Israel [note 1]

**DOI:** 10.3390/tropicalmed8060319

**Published:** 2023-06-13

**Authors:** Eyal Itzkowitz, Evan A. Alpert, Abdulhadi Z. Farojeh, Deena R. Zimmerman, Eli Schwartz, Tamar Lachish

**Affiliations:** 1Nephrology Unit, Shaare Zedek Medical Center, Jerusalem 9103102, Israel; itzeyal@szmc.org.il; 2Department of Emergency Medicine, Shaare Zedek Medical Center, Jerusalem 9103102, Israel; avraham.alpert@mail.huji.ac.il; 3Faculty of Medicine, Hebrew University of Jerusalem, Jerusalem 9101001, Israel; lachisht@szmc.org.il; 4TEREM Urgent Care Centers, Jerusalem 9439029, Israel; af@terem.com (A.Z.F.); dz@terem.com (D.R.Z.); 5Maternal Child and Adolescent Department, Public Health Division, Israel Ministry of Health, Jerusalem 9446724, Israel; 6The Center for Geographic Medicine and Tropical Diseases, The Chaim Sheba Medical Center, Tel Hashomer 5262000, Israel; 7Sackler School of Medicine, Tel-Aviv University, Tel-Aviv 6997801, Israel; 8The Infectious Diseases Unit, Shaare Zedek Medical Center, Jerusalem 9103102, Israel

**Keywords:** travel-related illness, travel medicine, low–middle-income countries, high-income countries, diarrhea, injuries

## Abstract

Information regarding post-travel morbidity is usually reported via dedicated post-travel clinics and mainly relates to travelers returning from low–middle-income countries (LMIC), however, the spectrum of morbidity seen within the community setting is scarcely reported. This prospective observational study among visitors to 17 community Urgent Care Centers (UCC) was designed to evaluate the reasons for post-travel community clinic visits and to compare travelers returning from LMIC to high-income countries (HIC). All visitors within one-month post-travel to all destinations were included. A total of 1580 post-travel visits were analyzed during 25 months. Travelers to LMICs were younger (mean 36.8 years old vs. 41.4 in the HIC group) and stayed longer periods abroad (30.1 ± 41.2 vs. 10.0 ± 10.6 in the HIC group) but more of them had pre-travel vaccines (35.5% vs. 6.6%). Travel-related morbidity was significantly more common in the LMIC group 58.3% (253/434) vs. 34.1% (391/1146) in the HIC group, (*p* < 0.001). Acute diarrhea was the leading cause of morbidity after visiting LMIC (28.8%) and was significantly more common than in the HIC (6.6%, *p* < 0.001). Other common morbidities in the LMIC cohort were respiratory (23.3%), cutaneous (15.8%), and injuries (9.9%). In the HIC group, the common morbidities were respiratory (37.3%), and diarrhea composed only 6.6% of the complaints. Our study group represents a less biased sample of travelers to LMIC as well as HIC, therefore, data from the UCC setting and at the specialized travel clinics complete each other in understanding the true extent of morbidity in travelers.

## 1. Introduction

International travel has grown significantly in recent decades, accounting for approximately 1.4 billion travelers annually [1]. Although the COVID pandemic caused a significant decline in worldwide travel [2,3,4], the growth in international travel is spiking again. Over the years, the profile of the passengers has also changed and includes an increasing number of children and the elderly, those with chronic illnesses, and an increased rate of travel to exotic destinations, including low–middle-income countries (LMIC) [5,6,7,8,9]. Information about travel-related morbidity is important for health practitioners to diagnose disease in returning travelers and for consulting those seeking pre-travel advice. This knowledge is relevant to the designated pre- and post-travel clinics, but also to a diverse range of other health specialties, such as primary care physicians or urgent care settings, as travelers also access these facilities.

The largest database for travel-related morbidity is GeoSentinel, which consists of information collected from dedicated travel clinics [10,11,12,13]. This information is often biased due to the nature of referrals to these designated clinics, which attract patients with specific or more severe illnesses.

There is no systematic data collection of referrals to Urgent Care Centers (UCC) after traveling abroad. UCC provide walk-in care for minor injuries and illnesses outside the primary care clinics or emergency departments. There are a range of services provided depending on the center [14]. This study hypothesizes that patients who are seen at UCC in Israel (which treat various medical problems and minor traumas and function as after-hours primary care clinics) represent a group with different problems than those seen in designated travel clinics. This collection of data will complement the picture of the morbidity seen after international travel.

The objective of this study was to evaluate the reasons for post-travel community clinic visits and to compare travelers returning from LMIC to HIC. In addition, we assessed the difference in the morbidity spectrum compared to that seen in designated post-travel clinics. To our knowledge, no such work has yet been published in the travel medicine literature.

## 2. Materials and Methods

### 2.1. Study Design

This is a prospective observational study, based on data collected from 17 UCCs throughout Israel, known as TEREM (acronym in Hebrew for Immediate Medical Treatment).

TEREM provides medical examination services, laboratory testing, and imaging. The centers treat various medical illnesses and injuries and refer only the more severe cases (~5% of all cases) to hospital emergency departments (ED).

The data of the medical sessions are recorded in a uniform electronic medical record. For this study, the TEREM computerized management software had been updated and adapted for the research questions.

### 2.2. Study Population and Data Collection

The study population included travelers of all ages that visited any one of the TEREM clinics within one month after their return from any destination abroad between 21 September 2016 to 16 October 2018.

Identifying patients who returned from abroad during the previous month was performed upon registration by the clerk at TEREM via a checkbox question in the electronic medical record (EMR). For patients who answered “yes”, a dialog box and additional checkboxes were opened in the EMR that required the triage nurse and the treating physician to complete details of the nature of the recent visit and travel abroad (Appendix A). The patient’s medical records were automatically collected in a dedicated database that also included demographics, the clinical course, and discharge diagnosis. The collected data were numerically encoded and transferred anonymously to an Excel spreadsheet (Redmond, WA, USA: Microsoft).

The study was approved by the Helsinki Committee (Institutional Review Board) of the Shaare-Zedek Medical Center (protocol code 0183-17-SZMC). The study was observational and non-interventional; therefore, a waiver was granted to no longer require written informed consent.

## 3. Definitions

### 3.1. Categorizing Travel Destinations

Travel destinations were divided into HIC and LMIC. The definition was based on World Development Indicators as published in the World Bank database [15].

### 3.2. The Link between Morbidity and Travel

All travelers reporting a visit abroad within one month of their clinic visit were evaluated. The study cohort was divided into travel-related illnesses and non-travel-related illnesses. The distinction between travel-related and non-travel-related morbidity was made first by the TEREM treating physician and confirmed by two travel medicine specialists who reviewed the charts, considering the clinical description, incubation periods, and laboratory results. A non-infectious but travel-related medical problem was defined as a new disease that occurred during travel abroad or was aggravated during travel.

The study cohort was divided according to the travel destination—LMIC versus HIC.

### 3.3. Statistical Analysis

The statistical analysis was performed using Microsoft Excel 2010 (Redmond, Washington, DC, USA). Descriptive statistics were used to present the demographic data of the study population. A two-tailed T-test was used to explain the difference between continuous demographic data. Proportions are described with mean, median, and range where appropriate. Continuous variables are described with mean and standard deviation. The Chi-square test was utilized to explore differences in the prevalence of dichotomous data.

## 4. Results

During the 26 months of the study period, a total of 1600 patients were enrolled in the TEREM clinics within one month after their return from international travel. Twenty patients were excluded due to missing data, resulting in 1580 travelers (97.5%) who were assessed. Figure 1 depicts the distribution of travelers who were included in the study according to their travel destination. Travelers to LMIC accounted for 27.5% (*n* = 434) of all returnees, among whom, 72 visited the Indian subcontinent. The rest of the travelers (*n* = 1146) returned from HIC. The characteristics of these two groups are presented in Table 1.

Travelers to LMIC were significantly younger (Table 1). The age distribution of the post-travel patients attending the centers is depicted in Figure 2. Travelers aged 20–30 were the dominant age group in both cohorts. Gender representation was equal between the two groups. Travelers to the Indian subcontinent (not presented separately in the table) were younger (average 34.3 years), with a male predominance (55.6%).

The mean duration of the trip to LMIC was three times longer than the travel duration to HIC (30.1 days vs. 10.0 days, *p* < 0.001).

Among those who attended TEREM clinics within one-month post-travel, the travel-related morbidity was significantly higher in those visiting LMIC compared to HIC, 58.3% versus 34.1%, (*p* < 0.001). Table 2 describes the spectrum of morbidities seen among these patients according to the travel destinations of both groups.

In the LMIC cohort, the major illnesses were acute diarrhea (28.9%), respiratory illness (23.3%), dermatologic disorders (15.8%), and injuries (9.9%). Systemic febrile illness accounted for only 2.8%. Among them, four cases were for suspected malaria after traveling to sub-Saharan Africa and three cases were for suspected dengue fever after traveling to Thailand.

In the HIC group, diarrheal diseases were significantly lower and accounted for only 6.6% of complaints (*p* < 0.001). Respiratory illnesses were the major concern in this group (37.3% vs. 23.3% in the LMIC group, *p* < 0.002).

In the sub-category of ill returning travelers from the Indian subcontinent (*n* = 54, a part of the LMIC study group, not presented separately in the tables), the most common reason to visit TEREM was acute diarrhea, which accounted for 61.1% of complaints and was significantly higher compared to other LMIC and HIC (*p* < 0.001), followed by cutaneous disease (11.1%) and respiratory illness (9.2%). Injuries were less frequent (5.5%).

Most injuries that occurred while traveling (*n* = 75, including both HIC and LMIC), were due to falls (53%). Road accidents accounted for 12% and sporting activities accounted for 7% of the injuries. There was no significant difference in the rate of injuries between the groups.

The rate of complaints related to chronic illnesses (part of the ‘Other’ group, Table 2) was low (0.8% of all travel-related complaints with no significant difference between LMIC to HIC travelers).

Altogether, most problems among the post-travel-related morbidities, were self-resolved or solved at the level of the primary or urgent care physician. Of the travel-related illnesses, 45 patients were referred to the ED for further tests (6.7% of patients from the LMIC group and 7.2% from the LMIC group, *p* = 0.94, Table 1). Of those referred to the ED, 23.4% were due to suspected cardiovascular disorders, such as pulmonary embolus or myocardial infarction.

In addition, the travelers were asked if they received vaccinations before the trip. Of those traveling to LMIC, 35.5% were vaccinated, while the rate was only 6.6% among those traveling to HIC. The rate of vaccine recipients in the Visiting Friends and Relatives (VFR) group (not presented in the tables) traveling to LMIC (*n* = 50) was 18% and was significantly lower than the average among non-VFR travelers to LMIC (37.8%, *p* = 0.006). Sub-Saharan Africa was visited by 76 travelers, including 12 in the category of VFR. Of the VFR, only two were vaccinated before travel (16.7%), compared to the rest of the sub-Saharan African travelers (*n* = 64), of whom 50% were vaccinated.

## 5. Discussion

The current study demonstrates that post-travel morbidities are seen in UCC as well as specialized travel clinics. Most of the complaints seen in these settings are minor or self-limiting. UCC are not designated post-travel clinics and, therefore, the strength of our current study is the ability to compare travelers returning from LMIC to travelers returning from HIC. The category of non-travel-related morbidity (59% of all returning travelers) emphasizes that UCC are a primary care setting.

Among patients attending the UCC within one month after return, a total incidence of 41% travel-related morbidity despite not being designated post-travel clinics is impressive. The incidence of travel-related morbidity was significantly higher upon returning from LMIC than upon returning from HIC (*p* < 0.001). The higher incidence of morbidity in the LMIC group occurred despite the backdrop of younger travelers that visited LMICs for longer periods and even though vaccination rate was higher in this group. The spectrum of diseases also differed according to the travel destination. Gastrointestinal illness ranked first in travelers returning from LMIC compared to respiratory illness upon return from HIC (Table 2). The mean days from return to arrival at TEREM Table 1) was very short in both groups, therefore, the chance for locally acquired infections (not travel-related) was negligible.

Comparing our LMIC group results to the designated large GeoSentinel registry, gastrointestinal complaints (acute and chronic) were the most common morbidities described after returning from LMIC countries, 34.8% in our study versus 42% in the GeoSentinel registry [10,11]. Gastrointestinal complaints were even more common in returnees from the Indian sub-continent (61.1%). These numbers are in proportion with further literature regarding relative rates of acquisition of gastrointestinal infections after traveling to LMIC, with the highest relative rates after visiting the Indian subcontinent [16,17].

On the other hand, respiratory illness (23.3%) ranked second in our study versus the being fourth most common morbidity seen by the GeoSentinel clinics—13% percent of cases. Cutaneous diseases were represented equally (15.8%). Systemic febrile illnesses after traveling to LMIC, which account for 22% of referrals to GeoSentinel travel clinics [10,11], were observed in only seven cases in our study group (2.8%). Of these, four cases were suspected of having malaria after traveling to sub-Saharan Africa and three more were suspected of dengue fever after traveling to Thailand (final diagnosis was not available due to lack of outcome data). The paucity of systemic fever cases in our study can be explained by the fact that, in these cases, the patients most likely referred directly to the ED or a designated post-travel clinic. Among travelers to HIC, no cases of systemic febrile illnesses were reported.

Regarding non-infectious diseases, which are less reported in the travel medicine literature, injuries ranked fourth in our current LMIC group (9.9%), while they are hardly seen by designated post-travel clinics. In the GeoSentinel, these account for only 1.4% of total referrals to travel clinics [10,11]. The difference is obvious, as the information collected by the GeoSentinel is biased towards more selected tropical diseases, while more severe cases and non-infectious diseases are less often recorded.

The spectrum of diseases seen in travelers to HIC is rarely described in the literature, and when reported, this is usually only from designated post-travel clinics [18]. In the HIC, the common morbidities were respiratory (37.3%) and diarrhea, which composed only 6.6% of the complaints.

Most of the traumas seen in this study were minor and occurred due to falls (53%). As some of the injuries were probably treated while being abroad and as the UCC do not usually treat major trauma, the injury rate and the severity among travelers to all destinations were probably substantially underestimated in our study. It is known that the incidence of fatal injuries in US citizens is higher abroad than at home [19,20]. A recent systematic review showed that overall, travelers appear to have a higher risk of mortality due to injury than among domestic populations [21], but the rate of non-fatal injuries among travelers is not known. Our study helps to complete the picture of travel-related injury by reducing the referral bias and adding the spectrum of trauma.

The rate of complaints related to exacerbations of chronic illnesses due to travel was surprisingly low—0.8% (part of the other group—Table 2). This group is also rarely represented in the GeoSentinel database. Further studies in the community care setting are needed to assess chronic comorbidities after return from travel abroad. This can be evaluated by assessing the changes needed in regular medication prescriptions and the rate of exacerbations of chronic disease related to travel. The extent of visits to medical facilities due to chronic illness during travel is also under-reported and should be better investigated [22,23].

Vaccine-preventable diseases are significant contributors to morbidity and potential mortality in travelers. Based on our questionnaire, of those traveling to LMIC, only 35.5% were vaccinated according to area-specific recommendations. There is no information on the true percentage of Israelis vaccinated before travel, but this disappointingly low number could be a rough estimation as they were seen in UCC throughout Israel without any known bias. The highest awareness of the need for pre-travel vaccination was among travelers to the Indian subcontinent (61.1%), yet this percentage is also far from optimal. The lowest compliance for pre-travel vaccination was observed among VFR (18%), similarly to the literature published in international databases (15%) [10,11,24], and significantly lower compared to the non-VFR population in our study traveling to LMIC destinations (37.8%). Similarly to the global literature among VFRs traveling to sub-Saharan Africa, an even lower rate of pre-travel vaccinations (16.6%) was found compared to the rest of the travelers to this destination (50%). VFRs to all destinations in LMIC continue to pose a challenge with their low compliance with pre-travel preventive advice.

## 6. Limitations

A substantial limitation is the small number of patients and that all were Israeli travelers. The small numbers limited some of the required comparisons. Yet the referral of all patients seeking medical care post-travel gives an unbiased evaluation of post-travel morbidity (and not only the more severe cases). A second limitation is that there are no outcome data. We do not know the final diagnosis of patients referred to the ED, culture results, etc. According to the nature of UCC, which are not designated to reach final diagnosis, the study looked at general syndromic categories of diagnosis, e.g., “dermatologic disorders”, thereby losing the specificity and robustness of a single diagnosis. In addition, this study may have missed the severe cases which were brought directly to the hospital. Nonetheless, the TEREM database has valuable data regarding mild comorbidities and minor illnesses, data that up to now have been scarce. A third limitation is that we could not calculate the mean time between return and clinical symptoms, therefore, there is a possibility that some of the infections were locally acquired post-travel. However, we did calculate the mean days from return to arrival at TEREM (Table 1) and this period was quite short (2.3 to 2.8 days), therefore the chance for locally acquired infections is negligible.

The results are limited to Israel and specifically to TEREM UCC. TEREM maintains a comprehensive electronic medical record and extensive laboratory and imaging capabilities, which may explain why returning travelers feel comfortable using these services. However, this may not be true with UCC with fewer facilities, or family practice clinics. A recent study found that there is a significantly lower percentage of referrals to the ED from TEREM centers compared to a group of UCC in the US [25]. Our results should also be assessed in other UCC settings and in other community settings, especially family practice clinics.

## 7. Conclusions

In conclusion, our study complements the literature by reporting the extent of minor morbidities and injuries which are seen in the community setting and are rarely reported by the designated travel clinics. As such, our study group represents a less biased sample of the population traveling to LMIC as well as HIC. Most of the illnesses seen in the UCC were self-limited or solved by the urgent care physician without referral to a specialized travel clinic or hospitalization. Our additional finding of the low percentage of pre-travel vaccination is concerning.

Together, the data from the UCC setting and at the specialized travel clinics complete each other in providing an understanding the true extent of morbidity in travelers. More studies are needed from primary care settings who see returning travelers to cover this gap in travel medicine.

## Figures and Tables

**Figure 1 tropicalmed-08-00319-f001:**
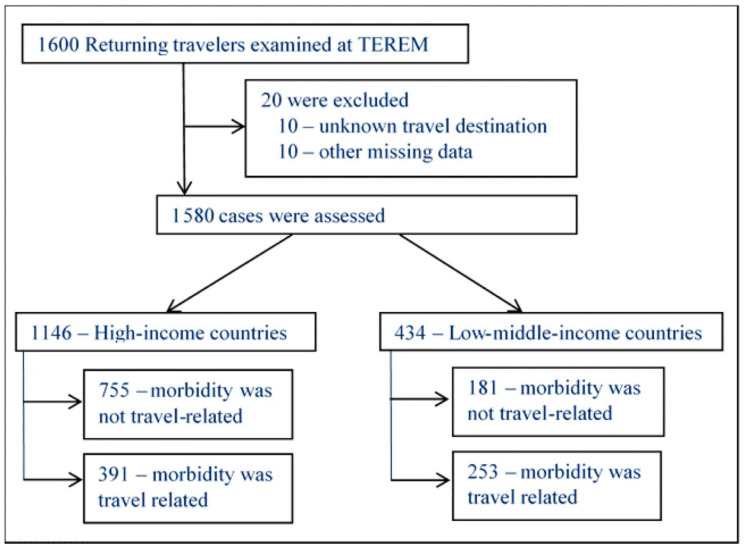
Distribution of travel-related illness and destination.

**Figure 2 tropicalmed-08-00319-f002:**
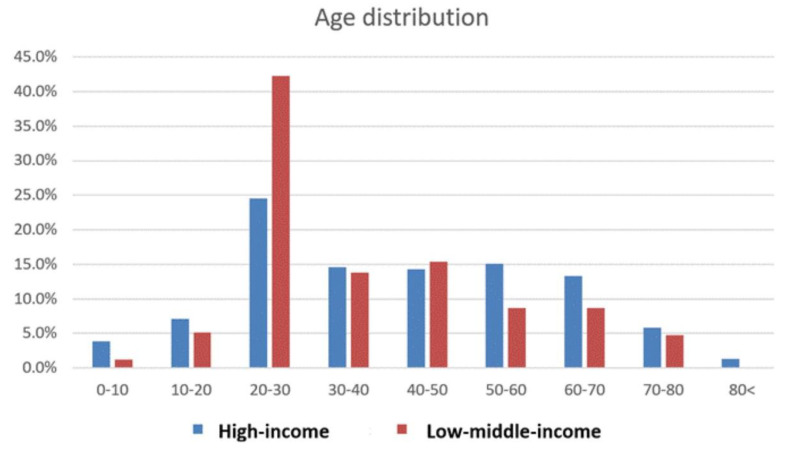
Age distribution of study population according to travel destination.

**Table 1 tropicalmed-08-00319-t001:** Characteristics of low–middle- vs. high-income countries.

	Low–Middle-Income Countries, *n* = 434	High-Income Countries, *n* = 1146	*p*-Value
Mean age (year ± SD)	36.8 (±16.4)	41.4 (±19.1)	*p* = 0.0017
Females, *n* (%)	218 (50.2)	578 (54)	*p* = 0.35
Pre-travel vaccinations, *n* (%)	154 (35.5)	76(6.6)	*p* < 0.001
Mean days from return to arrival at TEREM (±SD)	2.8 (±3.2)	2.3 (±3.6)	*p* = 0.16
Mean duration of trip (days ± SD)	30.1 (±41.2)	10.0 (±10.6)	*p* < 0.001
Referred to emergency department, *n* (%)	29 (6.7)	82 (7.2)	*p* = 0.94

**Table 2 tropicalmed-08-00319-t002:** Morbidities of low–middle- vs. high-income countries.

Morbidity	Low–Middle-Income Countries, *n* = 434 (%)	High-Income Countries, *n* = 1146 (%)	*p*-Value
Total of travel related morbidity	253 (58.3)	391 (34.1)	<0.001
Acute diarrhea	73 (28.85)	26 (6.6)	<0.001
Non-diarrheal gastrointestinal disorder	15 (5.9)	23 (5.9)	0.98
Respiratory illness	59 (23.3)	146 (37.3)	0.002
Dermatologic disorder	40 (15.8)	56 (14.3)	0.6
Injury	25 (9.9)	50 (12.8)	0.26
Cardiovascular disorder	9 (3.6)	24 (6.1)	0.14
Systemic febrile illness	7 (2.8)	0 (0.0)	0.009
Other *	25 (9.9)	66 (16.9)	0.01

* Other includes: musculoskeletal symptoms, nonspecific symptoms or signs, genitourinary disorder, sexually transmitted disease, ophthalmologic disorder, underlying chronic disease, lymphatic disorder, neurologic disorder, psychological disorder, chronic diarrhea, obstetrical or gynecologic disorder, dental problem, adverse drug or vaccine reaction.

## Data Availability

Data is unavailable due to privacy or ethical restrictions.

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
