# Peer review of "Morbidity of Returning Travelers Seen in Community Urgent Care Centers throughout Israel†"

_tropicalmed, 2023, doi:10.3390/tropicalmed8060319_

Round 1

Reviewer 1 Report

Thank you for your submission, it was a very interesting study to read. I have a couple of suggestions for you to consider:

Materials and Methods: I would suggest putting the statement about the IRB approval and the waiver for written informed consent under the Study Population and data collection section.

Results: Table 2: 

For the "total" row I would suggest  "Total of travel related morbidity".  It is confusing to see the header of n=434 but the first row states "total" with the number of 253- this way the reader knows the percentages are based on the n of 434 not the total number on the first column of 253.

Throughout the document please check for correct punctuations some have ., (line 219) and some are missing a period at the end of the sentence.

Your limitations and conclusions were accurate. I agree with you, more studies should be done on a larger scale to find the true extent of morbidity after travel.

References: All the references need to follow the AMA style (example is below). Please capitalize proper nouns such as GeoSential(reference # 12 & 13). Ireland needs to capitalized (#6) and Swiss needs to be capitalized (#8)

Reference #1 is missing the title of the website visited and the accessed date

All journal references should begin with the last name, not the first name initial. 

If you can add DOI numbers I would recommend that, several are missing page numbers as well. 

An example of AMA style for reference #2 is :

Klinger C, Burns J, Movsisyan A, Biallas R, Norris SL, Rabe JE, Stratil JM, Voss S, Wabnitz K, Rehfuess EA, Verboom B; CEOsys Consortium. Unintended health and societal consequences of international travel measures during the COVID-19 pandemic: a scoping review. J Travel Med. 2021 Oct 11;28(7):taab123. doi: 10.1093/jtm/taab123. PMID: 34369562; PMCID: PMC8436381.

I enjoyed reading your manuscript and look forward to seeing more studies like this.

Please review for punctuation and some grammar as identified in the comments on references.

Author Response

Thank you for your submission, it was a very interesting study to read. I have a couple of suggestions for you to consider:

Materials and Methods: I would suggest putting the statement about the IRB approval and the waiver for written informed consent under the Study Population and data collection section.

Done.

Results: Table 2: 

For the "total" row I would suggest  "Total of travel related morbidity".  It is confusing to see the header of n=434 but the first row states "total" with the number of 253- this way the reader knows the percentages are based on the n of 434 not the total number on the first column of 253.

Done.

Throughout the document please check for correct punctuations some have ., (line 219) and some are missing a period at the end of the sentence.

Done.

Your limitations and conclusions were accurate. I agree with you, more studies should be done on a larger scale to find the true extent of morbidity after travel.

References: All the references need to follow the AMA style (example is below). Please capitalize proper nouns such as GeoSential(reference # 12 & 13). Ireland needs to capitalized (#6) and Swiss needs to be capitalized (#8)

Reference #1 is missing the title of the website visited and the accessed date

All journal references should begin with the last name, not the first name initial. 

If you can add DOI numbers I would recommend that, several are missing page numbers as well. 

An example of AMA style for reference #2 is :

Klinger C, Burns J, Movsisyan A, Biallas R, Norris SL, Rabe JE, Stratil JM, Voss S, Wabnitz K, Rehfuess EA, Verboom B; CEOsys Consortium. Unintended health and societal consequences of international travel measures during the COVID-19 pandemic: a scoping review. J Travel Med. 2021 Oct 11;28(7):taab123. doi: 10.1093/jtm/taab123. PMID: 34369562; PMCID: PMC8436381.

The references were changed to AMA style. DOI was added.

I enjoyed reading your manuscript and look forward to seeing more studies like this.

Reviewer 2 Report

This study describes patients presenting in UCC (primary care) within one month after travel, and compares morbidity in HIC and LIC.

Awareness of travel related diseases is important (also in primary care) since urgent diagnosis and treatment is necessary for several travel related diseases (malaria, enteric fever,..)

However, about half of the morbidity reported in the patient population (755/1580) was categorised as not-travel-related, diminishing the relevance of the data. The issue of the large proportion of non-travel-related morbidity needs to be addressed.

Since morbidity was reported as syndromic categories, important information is lacking. I could imagine that the spectrum of disease in the respiratory disorder group in HIC is different from that in the LIC. Could the specific diagnoses/ aetiology be added in the data? This information is especially important for the respiratory disorder group since it accounts for the largest proportion of patient (in HIC).

Most patients presented a few days after return. What was the reason for the inclusion period of one month after return? Related to incubation periods? Other reasons?

Author Response

This study describes patients presenting in UCC (primary care) within one month after travel, and compares morbidity in HIC and LIC.

Awareness of travel related diseases is important (also in primary care) since urgent diagnosis and treatment is necessary for several travel related diseases (malaria, enteric fever,..)

However, about half of the morbidity reported in the patient population (755/1580) was categorised as not-travel-related, diminishing the relevance of the data. The issue of the large proportion of non-travel-related morbidity needs to be addressed.

The non-travel related category emphasizes that these are primary care clinics and not designated post-travel clinics. We added a sentence to address this issue in the first paragraph of the discussion and emphasized tha yet 41% was travel related (second paragraph).

Since morbidity was reported as syndromic categories, important information is lacking. I could imagine that the spectrum of disease in the respiratory disorder group in HIC is different from that in the LIC. Could the specific diagnoses/ aetiology be added in the data? This information is especially important for the respiratory disorder group since it accounts for the largest proportion of patient (in HIC).

As discussed in the limitations section the study looked at general syndromic categories of diagnosis and we do not have any outcome data. The reason for that is the nature of primary clinics that are not designated to reach final diagnosis. This was clarified better in the limitation section.

Most patients presented a few days after return. What was the reason for the inclusion period of one month after return? Related to incubation periods? Other reasons?

The only inclusion criteria was patients that visited any one of the TEREM clinics within one month after their return. This period was indeed chosen according to related incubation periods of 21-28 days.

Reviewer 3 Report

wonderful work

great results

i was a little intrigued by the respiratory illnesses higher in visitors to HIC countries. What is the explanation? it needs to be clearly discussed.

Acute GI infections common in travelers in both groups are fine but is the etiology different? did you try to look for it? is it possible to look for it?

what skin conditions were common in both the groups? is it infectious or non infectious?

One month is a long period of time for respiratory and GI infections to manifest after travel. How were the locally acquired infections during the time period between return and clinical presentation, ruled out?

Is there a possibility of calculating mean time between turn and clinical symptoms? can it be measured/ was it recorded? it might be interesting to know the time intervals in these cases.

Author Response

i was a little intrigued by the respiratory illnesses higher in visitors to HIC countries. What is the explanation? it needs to be clearly discussed.

Acute GI infections common in travelers in both groups are fine but is the etiology different? did you try to look for it? is it possible to look for it?

what skin conditions were common in both the groups? is it infectious or non infectious?

The questions are relevant and important but as discussed in the limitations section, the study looked at general syndromic categories of diagnosis and we do not have any outcome data. The reason for that is the nature of primary clinics that are not designated to reach final diagnosis. We hope our preliminary results will trigger further larger studies that will be able to collect final diagnosis.

One month is a long period of time for respiratory and GI infections to manifest after travel. How were the locally acquired infections during the time period between return and clinical presentation, ruled out?

As described in the methods section “The distinction between travel-related and non-travel-related morbidity was made first by the TEREM treating physician and confirmed by two travel medicine specialists who reviewed the charts, considering the clinical description, incubation periods, and laboratory results”.

Is there a possibility of calculating mean time between turn and clinical symptoms? can it be measured/ was it recorded? it might be interesting to know the time intervals in these cases.

We agree, but the data was not recorded standardized to allow us to calculate (some physicians wrote ‘several weeks/days…’, other wrote ‘a long period of…’, and only a few wrote an exact period of time).

Reviewer 4 Report

In this study the authors have compared the post-travel morbidities among Israeli travellers who returned from LMICs vs HICs. This is an interesting study and fills an important knowledge gap in this area (as we realise from the just completed ISTM congress in Basel). This is a potentially publishable paper and I suggest expediting its publication. I have a few minor comments for improvement.

1. This phrase ‘Urgent Care Centers’ in the title and elsewhere in the text is confusing, and can be replaced by another term used the text such as ‘Immediate Medical Treatment Center’ or ‘TEREM’ with its full spelling in Hebrew transliteration. ‘Urgent Care Centers’ is confusing with ER or A&E.

2. The abstract should also highlight that respiratory illnesses were more common among travellers from HICs. Also highlight in the abstract that travellers to LMICs were younger, stayed longer but more of them had pre-travel vaccines.

3. Use a consistent term like ‘respiratory illnesses’ and stick to that throughout the text, tables and figures. Using ‘respiratory illnesses’ and ‘respiratory disorder’ interchangeably is confusing.

4. Please remove Figure 3, it is repetitious of what’s presented in Table 2. 

5.  Please add ethics/IRB reference number.

6.       Discussion should be beefed up. Please discuss that the differences occurred in the backdrop of younger travellers visiting LMICs for a longer period even though vaccination rate was higher in them. Highlight that respiratory illnesses were more common among travellers from HICs. This is found in other settings. For instance influenza, meningococcal diseases and other infections that are transmitted via respiratory route equally affected Hajj pilgrims from LMICs and HICs (some bacterial infections were more common among travellers from HICs).

7.       Fixing of minor English punctuation issues is needed. For example, P1, line 29, “A total of 1580 post-travel visits that were analyzed during 25 months”, should be “A total of 1580 post-travel visits were analyzed during 25 months” (ie remove ‘that’). In the same page, line 34, “In the HIC the common morbidities were respiratory (37.3%)” should be “In the HIC GROUP the common morbidities were respiratory (37.3%)”. There are other examples elsewhere in the manuscript.

8. At times both percentage and numerator (n= xyz) have been reported but not at other times. For example, in this sentence, “In the LMIC cohort, the major illnesses were acute diarrhea (28.9%), respiratory illness (23.3%), dermatologic disorders (15.8%), and injuries (9.9%). Systemic febrile illness accounted for only 2.8% (n=7)” only for the last percentage (2.8%) numerator is shown, consistency is important. Also for brevity, you can refer to the table if the full data are there.   

9.      Please follow journals’ house style of referencing, and buttress your discussion by adding other references.

Finally, this is an interesting manuscript and can be accepted after minor revision.

English is generally good, fixing several minor punctuation errors is needed.

Author Response

  1. This phrase ‘Urgent Care Centers’ in the title and elsewhere in the text is confusing, and can be replaced by another term used the text such as ‘Immediate Medical Treatment Center’ or ‘TEREM’ with its full spelling in Hebrew transliteration. ‘Urgent Care Centers’ is confusing with ER or A&E.

We have maintained the consistency of Urgent Care Centers. Please see page 2 line 61 where the term “Clinics” was replaced with “Centers”. We also added the explanation: “Urgent Care Centers provide walk-in care for minor injuries and illnesses outside the primary care clinics or emergency departments. There is a range of services provided depending on the center.” The term Urgent care centers is well known in the Emergency Medicine literature and is consistent with the policy statement reference that we have added from the Annals of Emergency Medicine: “Urgent Care Centers. Ann Emerg Med. 2017 Jul;70(1):115-116. doi: 10.1016/j.annemergmed.2017.03.049. PMID: 28645399.”

  1. The abstract should also highlight that respiratory illnesses were more common among travellers from HICs. Also highlight in the abstract that travellers to LMICs were younger, stayed longer but more of them had pre-travel vaccines.

Done. See results section in the abstract.

  1. Use a consistent term like ‘respiratory illnesses’ and stick to that throughout the text, tables and figures. Using ‘respiratory illnesses’ and ‘respiratory disorder’ interchangeably is confusing.

Done throughout the text and tables. Thank you.

  1. Please remove Figure 3, it is repetitious of what’s presented in Table 2. 

Figure 3 was removed.

  1. Please add ethics/IRB reference number.

Done.

  1. Discussion should be beefed up. Please discuss that the differences occurred in the backdrop of younger travellers visiting LMICs for a longer period even though vaccination rate was higher in them. Highlight that respiratory illnesses were more common among travellers from HICs. This is found in other settings. For instance influenza, meningococcal diseases and other infections that are transmitted via respiratory route equally affected Hajj pilgrims from LMICs and HICs (some bacterial infections were more common among travellers from HICs).

Done – first paragraph of the discussion and page 7 line 242-244 . We did not add a remark regarding the hajj morbidities, as all our travelers originated from HIC but went to either HIC or LMIC. The Hajj pilgrims originate either from HIC or LMIC but arrive to a LMIC destination.

  1. Fixing of minor English punctuation issues is needed. For example, P1, line 29, “A total of 1580 post-travel visits that were analyzed during 25 months”, should be “A total of 1580 post-travel visits were analyzed during 25 months” (ie remove ‘that’). In the same page, line 34, “In the HIC the common morbidities were respiratory (37.3%)” should be “In the HIC GROUP the common morbidities were respiratory (37.3%)”. There are other examples elsewhere in the manuscript.

Done.

  1. At times both percentage and numerator (n= xyz) have been reported but not at other times. For example, in this sentence, “In the LMIC cohort, the major illnesses were acute diarrhea (28.9%), respiratory illness (23.3%), dermatologic disorders (15.8%), and injuries (9.9%). Systemic febrile illness accounted for only 2.8% (n=7)” only for the last percentage (2.8%) numerator is shown, consistency is important. Also for brevity, you can refer to the table if the full data are there.  

Done. 

  1. Please follow journals’ house style of referencing, and buttress your discussion by adding other references.

Done.

Finally, this is an interesting manuscript and can be accepted after minor revision.

 Thank you.

Round 2

Reviewer 2 Report

Thank you for these clarifications

Author Response

Thank you for these clarifications

Reviewer 3 Report

It is a very interesting attempt to analyse the data, but, there are far too many loose end in this research work.

yes, it raises more questions than it answers.

i think the data should be re analysed with details as asked for. this would probably be more useful. 

this data is more important for the local consumption. 

Author Response

It is a very interesting attempt to analyse the data, but, there are far too many loose end in this research work.

yes, it raises more questions than it answers.

i think the data should be re analysed with details as asked for. this would probably be more useful. 

this data is more important for the local consumption.

Thank you for reviewing the paper again. We cannot Calculate the mean time between turn and clinical symptoms, but in table 1 you can see the calculation of mean days from return to arrival at TEREM. The period is quite short in both groups therefore the chance for locally acquired infections is negligible. We added a sentence with this remark (end of the 2nd paragraph in the discussion section). However, as we explained previously the nature of primary clinics that are not designated to reach final diagnosis does not allow us to get more details and reanalyze.